

# Genetic structure of the threatened West-Pannonian population of Great Bustard (*Otis tarda*)

Jose L. Horreo[1], Rainer Raab[2], Péter Spakovszky[2,3] and Juan Carlos Alonso[4]

[1] Ecology and Evolution, University of Lausanne, Lausanne, Switzerland
[2] Technisches Büro für Biologie, Deutsch-Wagram, Austria
[3] Institute of Wildlife Management and Vertebrate Zoology, University of West Hungary, Sopron, Hungary
[4] Evolutionary Ecology, Museo Nacional de Ciencias Naturales (CSIC), Madrid, Spain

## ABSTRACT

The genetic diversity, population structure and gene flow of the Great Bustards (*Otis tarda*) living in Austria-Slovakia-West Hungary (West-Pannonian region), one of the few populations of this globally threatened species that survives across the Palaearctic, has been assessed for the first time in this study. Fourteen recently developed microsatellite loci identified one single population in the study area, with high values of genetic diversity and gene flow between two different genetic subunits. One of these subunits (Heideboden) was recognized as a priority for conservation, as it could be crucial to maintain connectivity with the central Hungarian population and thus contribute to keeping contemporary genetic diversity. Current conservation efforts have been successful in saving this threatened population from extinction two decades ago, and should continue to guarantee its future survival.

## INTRODUCTION

The Great Bustard (*Otis tarda*, Linnaeus 1758) is a globally endangered species, classified as vulnerable in the Red List of Threatened Species (*IUCN, 2015*). Today the stronghold of the species is found in the Iberian Peninsula, with ca. 70% of the world total abundance. Other central European and Asian populations are much smaller and show a fragmented distribution pattern (*Alonso & Palacin, 2010*). The German population and the West-Pannonian population living in Austria-Slovakia-West Hungary suffered particularly dramatic decreases, respectively from 4,100 birds in 1939 to 65 in 1995, and from ca. 3,500 birds in 1900 to 130 birds in 1996 (*Raab et al., 2010*; http://www.grosstrappe.at; http://www.grosstrappe.de). However, both populations survived and have even slightly increased in numbers, reaching nowadays 197 birds in Germany and 505 birds in the West-Pannonian population (February 2015 in both cases), thanks to continued and intensive conservation efforts, centered on habitat management programs in both countries plus captive breeding in Germany (*Langgemach & Litzbarski, 2005*; *Raab et al., 2010*; *Langgemach, 2012*; http://www.grosstrappe.at; http://www.grosstrappe.de).

Corresponding author
Jose L. Horreo,
horreojose@gmail.com

Specifically for the study population in the West-Pannonian region, conservation efforts since 2001 have centered on implementing large-scale agri-environmental measures funded through Austrian and Hungarian Agri-environmental Scheme for Great Bustard and Austrian, Hungarian and Slovakian Great Bustard LIFE, LEADER, INTERREG and Rural Development projects to (1) improve the quality of the habitat, (2) increase the species' reproductive success (details in *Raab, 2013*; http://www.grosstrappe.at, *Faragó, Sapkovsky & Raab, 2014*; *Raab et al., 2014a*; *Raab et al., 2014b*), and (3) reduce power line collision casualties, by burying powerlines or installing bird collision diverters (*Raab et al., 2012*). In spite of all these conservation efforts, the marked decreases in the West-Pannonian population and in Germany in just a few decades might have caused already a loss of the original genetic diversity and an increase in inbreeding in these populations.

Genetic analyses have been done in this species not only for its phylogenetic study (*Pitra et al., 2002*; *Arif et al., 2012*; *Horreo, Alonso & Mila, 2014*), but also for conservation purposes. These studies have shown that Iberian and central European populations remain two distinct evolutionary significant units (*Pitra, Lieckfeldt & Alonso, 2000*) despite their considerable dispersal and migratory capacity. Moreover, significant genetic differentiation was suggested among local populations within these two subpopulations, highlighting the need to conserve not only the Iberian stronghold, but also all other smaller extant breeding groups of central Europe and Asia, in order to preserve the current genetic pool of the species. An effective conservation program needs a detailed knowledge of the genetic structure of the different subpopulations, but such studies have only been published for the Iberian Peninsula and Morocco (*Martin et al., 2001*; *Broderick et al., 2003*; *Alonso et al., 2009*; *Horreo et al., 2014*), with only limited data for other European populations (*Pitra et al., 1996*; *Pitra, Lieckfeldt & Alonso, 2000*).

The aim of this work is to explore for the first time the population structure, genetic diversity, and gene flow in the Austrian-Slovakian-West Hungarian (known as West-Pannonian) population of Great Bustards, using fourteen recently developed microsatellite loci (*Horreo et al., 2013*). The results should be useful not only for the conservation of this isolated central European population, but also to preserve the genetic diversity of this globally endangered species.

## MATERIALS AND METHODS

During the 2012 and 2013 breeding seasons (between early April and early September), Great Bustard moulted feathers were searched as DNA source in all areas previously known to be used by the species in Austria. In Austria, the authors had verbal permission from hunters to collect feathers in their properties. No other official permit is needed in this country to collect feathers. As for Hungary, the authors had a permit to collect gene samples issued for the Hungarian LIFE project (LIFE04 NAT/HU/000109), in which the Hungarian Ministry of Environment and Water was a co-financier and the regional competent Fertő–Hanság National Park Directorate was a partner. Single or small groups of feathers were found at 80 sites, covering the three most important breeding areas in Austria: Weinviertel, Marchfeld and Heideboden (Fig. 1). To minimize the probability
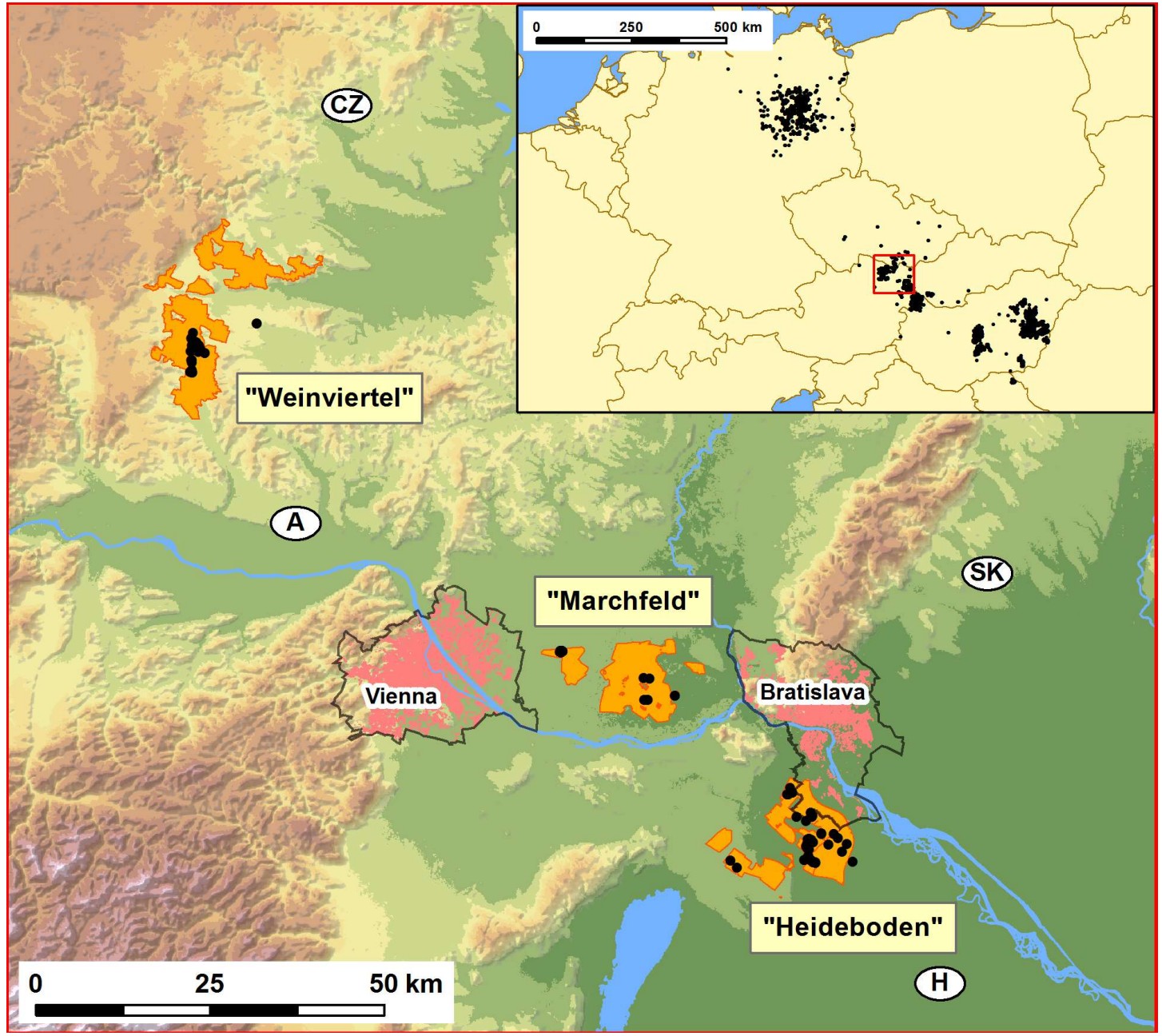

**Figure 1** **Map showing the distribution of the studied Great Bustard breeding areas in Austria-West Hungary-Slovakia (Weinviertel, March-feld and Heideboden; orange patches), and the location of the study area within central Europe (inset map on the upper right corner).** These three orange areas include, respectively: SPA "Westliches Weinviertel" (AT1209000), ca.7441 ha; a large part of SPA "Sandboden und Praterter-rasse" (AT1213V00), ca. 11.083 ha; and SPA "Parndorfer Platte—Heideboden" (AT1125129) -ca. 7260 ha- plus the northern part of Hungarian SPA "Mosni-sík" (HUFH10004) -ca. 3159 ha- and the Slovakian SPA "Sysl'ovské polia" (SKCHVU029) -ca. 1777 ha-. The black dots indicate the collection sites of the samples used for genetic analyses. The two capital cities, Vienna and Bratislava (pink patches), the perimeters of the suburban areas around them (black lines around pink patches), and the initials of the four countries converging in this geographic region (Austria, Czech Republic, Slovakia and Hungary), and their national borders (black lines) are also indicated. The inset map shows the distribution in central Europe in the period 1995–2014 (Source: data base from Rainer Raab, including information from Péter Spakovszky and Miklós Lóránt for Hungary and Torsten Langgemach and Henrik Watzke for Germany).

that two of the feathers collected for genetic analysis were of the same individual we used only a single feather from each group of moulted feathers found at each of these sites. Another 18 tissue and feather samples from the collections of the Vienna Natural History Museum (years 2000–2006) were used, totalling 98 individuals. DNA was extracted from the 80 feather samples and fourteen Great Bustard microsatellite loci were amplified three independent times following protocols published in *Horreo et al. (2014)*. Additionally, museum DNA samples were checked for DNA quality before amplification and their PCRs were done in laminar flow cabinets. These museum samples did not affect our analyses since the species' longevity is estimated at 10–15 years (JC Alonso, 2016, unpublished data). Introgressive laser signals from one channel to another during genotyping were carefully controlled. In order to avoid resampling of individuals, the genetic Identity Analysis of the Cervus software v.3.0 (*Kalinowski, Taper & Marshall, 2007*) was carried out.

Microchecker was employed to search for null alleles, scoring errors and large allele dropout (*Van Oosterhout et al., 2004*). Linkage disequilibrium was tested with Genepop on line software (http://genepop.curtin.edu.au). Deviations from Hardy–Weinberg equilibrium after Bonferroni correction (HWE), the number of alleles per locus (Na), the expected and observed heterozygosities averaged across loci (respectively, $H_S$ and $H_O$), the inbreeding coefficient ($F_{IS}$) and the Fixation Indexes ($F_{ST}$) and their associated *P*-values, were calculated with GenoDive 2.0b25 (*Merimans & Van Tienderen, 2004*). Allele frequency distribution was plotted with GenAlex v.6.5 (*Peakall & Smousse, 2012*). Allelic Richness (AR) was estimated with FSTAT v.2.9.3.2 (*Goudet, 1995*). The presence of past genetic bottlenecks was tested with the software Bottleneck v.1.2.02 (*Piry, Luikart & Cornuet, 1999*), with the most suitable settings for microsatellite loci data: the two-phase model (TPM) assumptions estimating *P*-values with the Wilcoxon signed-rank test (10,000 interactions).

Several different approaches were used to ensure robust quantification of the population differentiation in the study area. The Bayesian clustering method and Markov Chain Monte Carlo (MCMC) simulation implemented in STRUCTURE 2.3.4 (*Pritchard, Stephens & Donnelly, 2000*) were used to assess population structure. The STRUCTURE analyses were run by using an admixture model and correlated allele frequencies with a burn-in period of 50,000 replicates and sampling period of 500,000 replicates for number of clusters ($K$) from 1 to 3. Ten independent runs were performed for each $K$. We run two sets of STRUCTURE analyses: one without using sample location as prior, one using sample location as prior (LocPrior option). To determine the number of genetic clusters ($K$), we used the DK method of *Evanno, Regnaut & Goudet (2005)* based on the second order rate of change in log Pr $(X|K)$ as implemented by the program Structure Harvester v.0.6.94 (*Earl & VonHoldt, 2012*). The second approach was BAPS v.6.0 (*Corander et al., 2008*), which given a maximum value of partitions, uses a stochastic optimization procedure to find the clustering solution with the highest marginal likelihood of $K$ (i.e., an approximation of the most probable number of differentiated genetic populations conditional on observed data). Settings: admixture of individuals based on mixture clustering after clustering of groups of individuals; 1,000 iterations used to estimate the

admixture coefficients for the individuals; 200 reference individuals from each population (as recommended by the software developers); 10 genetically diverged maximum groups used to estimate the admixture coefficients for the reference individuals. The third approach was a discriminant analysis of principal components (DAPC), a multivariate method implemented in the *adegenet* package (v. 1.3-1) for the *R* software (*Jombart, 2008*) that identifies clusters of individuals without using any population genetic model. We used it in two different ways: (1) using the *find.clusters* function for the identification of the optimal *K* with the *choose.n.clust* option and the Bayesian Information Criterion (BIC); after that, DAPC was employed to assign individuals into populations, retaining the n/3 number of principal components (as recommended in the manual) with 80% of the cumulative deviance (which removes the effect of assigning populations a priori on the eventual assignment to clusters and offers an unbiased interpretation of population structure); and (2) without searching and optimal *K*: each individual was a priori assigned to its location of origin, obtaining for each individual the probability of assignment to their populations of samplings. Lastly, Oncor (*Kalinowski, Manlove & Taper, 2008*) Leave-One-Out test assignments (default settings) were employed also for searching/-confirming population structure and for confirming Structure/BAPS/adegenet results. These analyses delete each individual of the total dataset and try to assign it to one of the given populations; results show percentages of correct assignations for each population. The effective sizes ($N_e$) of the genetic estimated units were calculated with LDNe v.1.31 software (*Waples & Do, 2008*) employing random mating and 0.05 lowest allele frequency settings.

Contemporary gene flow among populations was studied under the assignment test criterion of *Rannala & Mountain (1997)* with Geneclass 2.0 (*Piry et al., 2004*), which detect individuals with immigration ancestry of up to two generations ago. The numbers of migrants per generation ($N$m) among breeding areas were estimated by the mean frequency of private alleles with the above-mentioned Genepop software.

# RESULTS

Two of the fourteen available microsatellite loci were discarded from analyses for different reasons: the Ot3 locus did not amplify (probably due to mutation(s) in the primer(s) area(s)) and the Ot6 locus was monomorphic in the dataset. Within the other twelve loci, amplification success was 87.4% and Microchecker discarded the presence of scoring errors, large allele dropout and null alleles in the dataset. The Identity Analysis showed two individuals with the same genotype in the Weinviertel breeding area, one of which was deleted in order to avoid resampling. The genetic variability (Table 1; Fig. 2) was high: the mean number of alleles per locus ranged between 3.25 (Marchfeld) and 5.17 (Heide-boden), and the allelic richness ranged between 3.07 (Marchfeld) and 3.60 (Heideboden). The smallest observed and expected heterozygosities were found in Heideboden (0.42 and 0.49, respectively), and the highest in Marchfeld (0.50 and 0.52, respectively). The inbreeding coefficients ranged between 0.03 in Marchfeld and 0.13 in the Heideboden, being positive in all three breeding areas, and no genetic bottlenecks were found in any

Horreo et al. (2016), *PeerJ*, DOI 10.7717/peerj.1759

**Table 1** Number of alleles per locus and outcome of tests for deviation from Hardy-Weinberg proportions (*P < 0.05, and **P < 0.01) after Bonferroni correction in the three sampled West-Pannonian Great Bustard breeding areas (see Fig. 1) for the 12 studied microsatellite loci.

| | Ot1 | Ot2 | Ot4 | Ot5 | Ot7 | Ot8 | Ot9 | Ot10 | Ot11 | Ot12 | Ot13 | Ot14 | Na | AR | $H_O$ | $H_E$ | $F_{IS}$ | B |
|---|---|---|---|---|---|---|---|---|---|---|---|---|---|---|---|---|---|---|
| Weinviertel $n = 35$ | 3** | 2 | 4 | 6 | 4** | 2 | 6 | 2 | 6** | 2** | 5 | 6 | 4.00 (1.76) | 3.27 (1.37) | 0.44 (0.22) | 0.49 (0.23) | 0.11 | 0.67 |
| Marchfeld $n = 14$ | 2 | 2 | 3 | 5 | 2 | 2** | 4* | 3 | 4 | 2** | 5 | 5 | 3.25 (1.29) | 3.07 (1.22) | 0.50 (0.28) | 0.52 (0.23) | 0.03 | 0.21 |
| Heideboden $n = 48$ | 3** | 2 | 6 | 9 | 5** | 1 | 6** | 4** | 7** | 3** | 10 | 6 | 5.17 (2.72) | 3.60 (1.72) | 0.42 (0.29) | 0.49 (0.29) | 0.13 | 0.27 |

**Notes.**

Acronyms: Na, mean allele number per locus; AR, allelic richness; $H_O$, observed heterozygosity; $H_E$, expected heterozygosity; $F_{IS}$, inbreeding coefficient; B, *P*-value for Bottleneck analyses..
Standard deviation is shown between brackets.

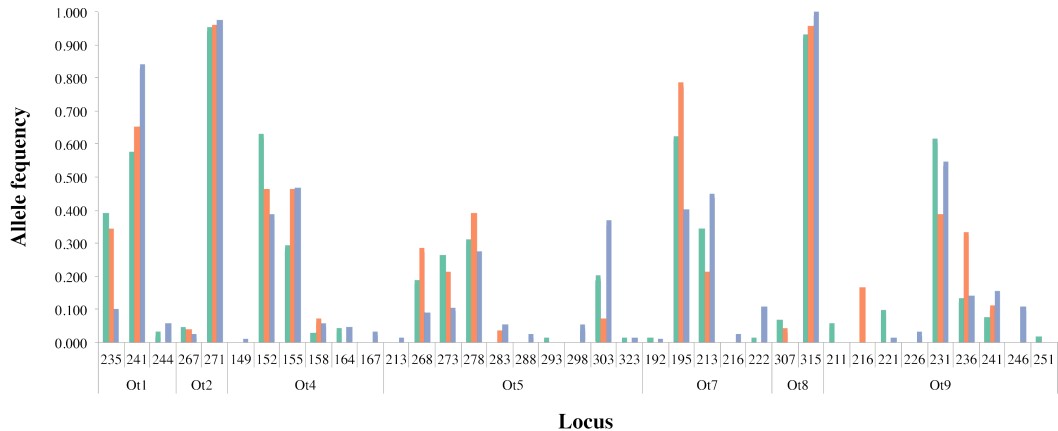

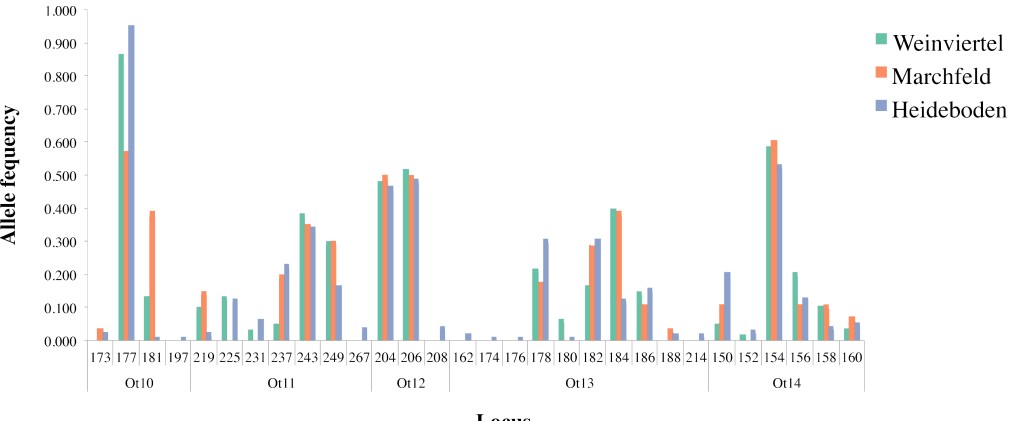

**Figure 2** Allele frequency distribution of the twelve analysed microsatellite loci across the three breeding areas.

of them. Non-significant deviations of HWE were found in 8, 9 and 5 out of 12 cases for Weinviertel, Marchfeld and Heideboden, respectively. Deviations from HWE were caused by heterozygote deficit. The Ot12 microsatellite locus showed significant HWE deviations in all the three breeding areas, so all analyses were done including and excluding it, with identical results in both cases (only results with this microsatellite locus are shown in the manuscript). Linkage disequilibrium was only statistically significant in a very small proportion of tests (3.03%).

According to Structure Harvester, Structure results (with and without LocPrior option) did not reveal a clear population structure across the study area, showing a unique genetic unit ($K = 1$) in the whole dataset. Even forcing $K = 2$ or 3, results did not show differences among breeding areas, they only showed mixed genotypes across all the three sampling points. Despite this, BAPS analyses showed the presence of two genetic subunits ($K = 2$; Fig. 3A) in the dataset, one composed by Weinviertel and Marchfeld, and another including the Heideboden individuals. DAPC analyses with the *find.clusters* function also revealed $K = 2$, but membership results showed no correspondence between genetic and spatial structures/populations (Fig. 3B), indicating the existence of a unique genetic unit

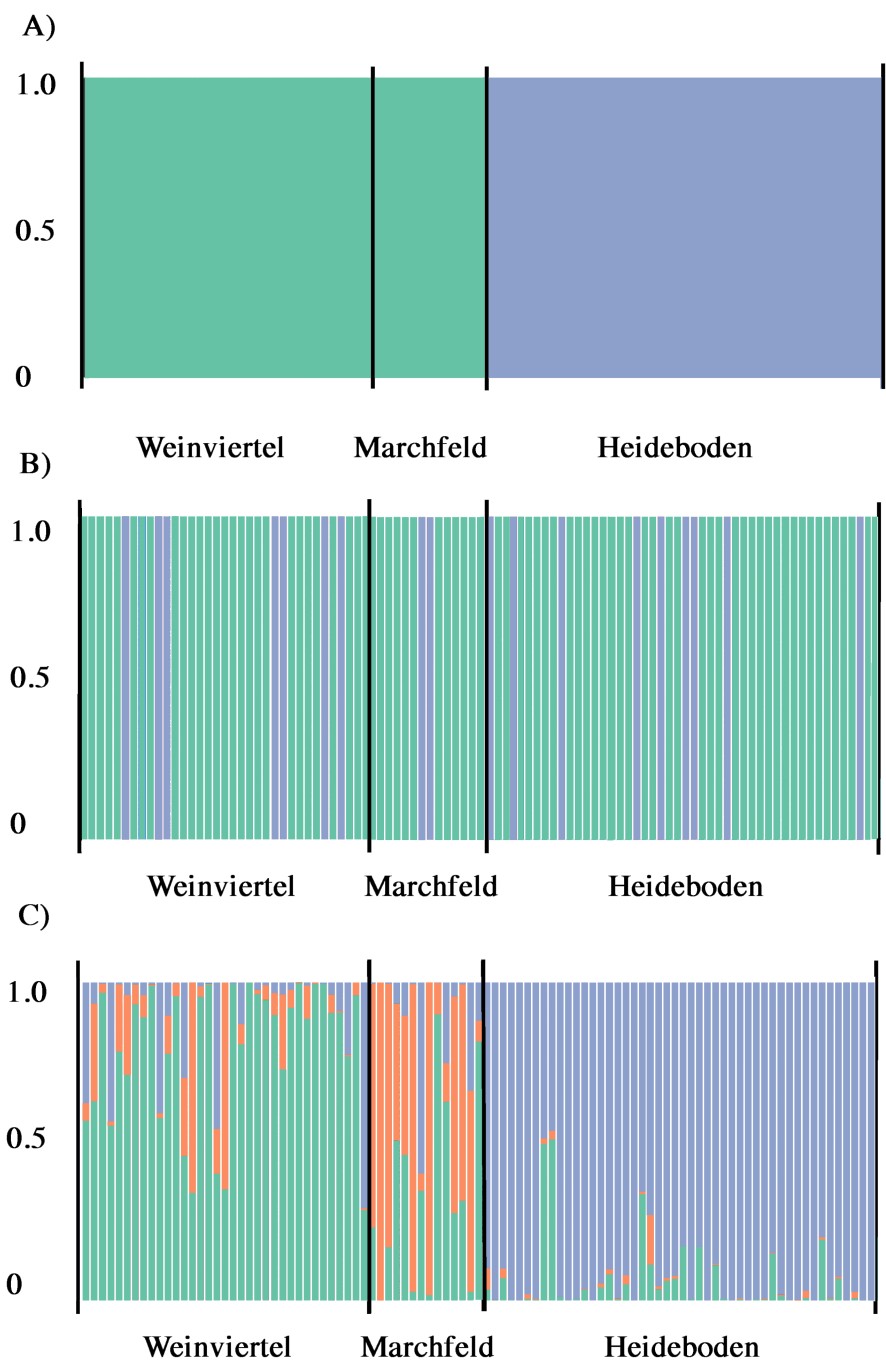

**Figure 3** Output of BAPS (A) and DAPC analysis with (B) and without (C) the *find.clusters* function showing the membership fraction (columns) of the inferred genetic units for West-Pannonian Great Bustards in each individual of the three studied breeding areas. Same colour in different individuals indicates that they belong to the same cluster.

**Table 2  Genetic differentiation among West-Pannonian bustard breeding areas pairs measured with $F_{ST}$ values (below diagonal; \*\*P-value < 0.01) and gene flow among them measured as the number of migrants per generation ($N$m; above diagonal).**

| $F_{ST}$/$N$m | Weinviertel | Marchfeld | Heideboden |
|---|---|---|---|
| Weinviertel | – | 2.39 | 3.07 |
| Marchfeld | 0.014 | – | 2.59 |
| Heideboden | 0.036\*\* | 0.055\*\* | – |

(as STRUCTURE suggested). When each individual was a priori assigned to its location of origin, obtaining for all individuals the probability of assignment to the populations where they were sampled, more similar results to the BAPS ones were found (Fig. 3C). This could mean that $K = 2$ is a biased interpretation of population structure. However, Oncor Leave-One-Out tests showed that when the three breeding areas were employed as independent areas, percentages of correct individual assignments were 64.7, 71.4 and 85.7% in Weinviertel, Marchfeld and Heideboden, respectively, whereas employing two subunits (following BAPS results) yielded higher assignment values: 95.8% of correct individual assignment for the genetic subunit composed by Weinviertel and Marchfeld, and 85.7% of correct individual assignment for the other genetic subunit (Heideboden). In addition to this, and in accordance with BAPS and Oncor results, $F_{ST}$ $P$-values were not significant between Weinviertel and Marchfeld and highly significant ($P$-val < 0.01) between Heideboden and the other two breeding areas (Table 2). For all these reasons, the effective population sizes ($N_e$) were estimated for the two mentioned genetic subunits, resulting in 30.3 individuals (95% CI [18.4–58.1]) for the subunit composed by Weinviertel and Marchfeld and 57.1 individuals (95% CI [28.5–247.9]) for the other subunit (Heideboden).

   According to the assignment tests on recent gene flow, 66.3% of the individuals were residents in the breeding area where they had been sampled when comparing the three sampling sites, and 81.6% when comparing the two genetic subunits (Heideboden *versus* Weinviertel plus Marchfeld). The number of migrants per generation ($N$m; Table 2) was high in all cases ($N$m > 2), being higher between Heideboden and the other two breeding areas (Weinviertel and Marchfeld) than between the two latter. The $N$m value between Heideboden and the genetic subunit including Weinviertel and Marchfeld individuals was 3.62, the highest of all estimated $N$m values.

## DISCUSSION

Our results show that the West-Pannonian population of Great Bustards displays high genetic variability (Na = 3.25–5.17, AR = 3.07–3.60, He = 0.49–0.52) in spite of a strong decrease in numbers of individuals suffered in the past decades, with a slightly smaller number of alleles per locus but higher allelic richness than the world's largest populations of this species (Na = 4.5–5.9, AR = 1.47–1.59, He = 0.50–0.59 without Ot3 and Ot6; *Horreo, Alonso & Mila, 2014* and JC Alonso, 2016, unpublished data). This, together with the results showing absence of genetic bottlenecks in this area, suggests that the marked

demographic decline suffered by West-Pannonian Great Bustards during the last century has not affected them in this respect, partly because the intensive protection measures probably stopped early enough the population decline (*Raab et al., 2010*). Overall, the genetic variability of the West-Pannonian population will probably be useful for avoiding major genetic changes (as genetic drift) in the near future and thus for maintaining the favourable trend of this population, provided that current habitat structure, climate conditions, management practices and other factors affecting this population remain constant in the future.

The West-Pannonian population showed also to be genetically sub-structured despite the relatively small area it occupies (Fig. 1). All population structure analyses used revealed the presence of a single population structure across the study area, with a hidden subpopulation structure (as found in different geographic regions in other bustard species; *Riou et al., 2012*): the birds in Heideboden (the largest, southern breeding group) were genetically different from the birds in other groups, forming a genetic subunit on their own, while the other two breeding areas (Weinviertel and Marchfeld) constitute together a second genetic subunit with a smaller effective size. In addition to this, subdivision into units with distinct gene frequencies creates a heterozygote deficit, so deviations from HWE in our dataset, caused precisely by heterozygote deficit, reinforce the existence of this sub-population structure within the entire population.

The above-mentioned genetic differences cannot be attributed to physical barriers since there are no such geographic obstacles between Heideboden and the other two breeding areas (Fig. 1). As in other bustard species (e.g., *Idaghdour et al., 2004*), contemporary gene flow (up to two generations) was detected in the dataset, and $N$m values were higher than one migrant per generation (OMPG; *Mills & Allendorf, 1996*) in all cases, including both gene flow among breeding areas and between the two genetic subunits. The gene flow inferred from high $N$m values and low genetic differentiation can be the result of a relatively recent demographic separation, in numbers of generations after the split. Despite this, the three breeding areas are separated by distances that can easily be covered by dispersing juveniles (*Martín et al., 2008*), and an exchange between the different Great Bustard breeding areas within the study area was indeed suggested based on local field observations of bird movements (*Raab, 2013*) as one of the possible causes for the rapid recovery of this population (*Raab et al., 2010*). These gene flow values ($N$m = 2.09–3.27) were also in accordance with those found for the species in other areas ($N$m = 2.0–7.6 within genetic units in Spain; *Horreo, Alonso & Mila, 2014*). Genetic differentiation and gene flow due to recent demographic isolation/gene flow barriers could be therefore discarded. The results on structure and gene flow patterns lead us to propose the whole area as a unique Great Bustard population conformed by two genetic subunits.

In addition to this, the whole West Pannonian population is not far from other Hungarian breeding groups, from which considering again the dispersal capacity of the species (*Martín et al., 2008*), dispersing immature individuals could have arrived and established at any of the breeding groups in Austria as breeding adults, contributing to keep the genetic diversity high. In particular, this could be the case in Heideboden, which is the West-Pannonian breeding aggregation closest to the Hungarian groups. Our

study showed that Heideboden is included within the whole population, but genetically differs from the other two Austrian subunits, having the highest genetic variability and effective size. This subunit, the largest among all breeding groups in the study area, seems to be not only the connection among central Hungarian and Austrian populations, but also a possible source of individuals for the other West-Pannonian genetic subunit ($N$m between Heideboden and the other Austrian subunit was higher than $N$m among breeding areas). These hypothesised movements could be confirmed through marking individuals. Therefore, within the conservation and management plan for the future, all extant breeding groups should be protected, but paying special attention to the Heideboden subunit, since due to its proximity to central Hungarian populations it could be crucial. This subunit should be therefore a priority for the conservation of other West-Pannonian Great Bustard breeding groups, and to guarantee their connection to the main Hungarian populations further east.

Summing up, our study is the first description of the genetic diversity and gene flow of the West-Pannonian population of Great Bustards based on analysis of microsatellite loci. This is one of the few populations that survived the sharp decreases that decimated this globally endangered species during the last century in central Europe. Our results identified one population in the study area, formed by two genetic subunits with high migration rates among them. The current genetic variability is high. Conservation efforts should be directed not only to preserve all extant breeding groups, but also the connectivity among them and with central Hungarian populations, in order to keep the genetic diversity of West-Pannonian Great Bustards at a high level. Current management and protection actions should therefore be maintained in the study area in order to guarantee the survival of this Great Bustard population.

## ACKNOWLEDGEMENTS

We thank Anita Gamauf for samples from the NHM Vienna, Victoria Antonaya for helping with laboratory tasks, Joaquín Calatayud for helping with *R* analysis, and Rita Castilho (PeerJ academic editor) and two anonymous referees for improving the quality of the manuscript.

### Funding

JLH was supported by a Marie Curie-Clarín CoFund grant (ACA14-26). Partial funding was provided by project CGL201236345 from the Spanish Directorate General for Scientific Research. Most feathers were collected within the three LIFE Projects "Crossborder Protection of the Great Bustard in Austria" (LIFE05 NAT/A/000077, www.grosstrappe. at), "Conservation of *Otis tarda* in Hungary" (LIFE04 NAT/HU/000109, www.tuzok.hu) and "Crossborder Protection of the Great Bustard in Austria—continuation" (LIFE09 NAT/AT/000225, www.grosstrappe.at. All three LIFE Projects are supported by the EU, many project partners and co-financiers. The funders had no role in study design, data collection and analysis, decision to publish, or preparation of the manuscript.

## Grant Disclosures

The following grant information was disclosed by the authors:

Marie Curie-Clarín CoFund: ACA14-26.

Spanish Directorate General for Scientific Research: CGL201236345.

Crossborder Protection of the Great Bustard in Austria: LIFE05 NAT/A/000077.

Conservation of *Otis tarda* in Hungary: LIFE04 NAT/HU/000109.

Crossborder Protection of the Great Bustard in Austria—continuation:
LIFE09 NAT/AT/000225.

## Competing Interests

The authors declare there are no competing interests.

## Author Contributions

- Jose L. Horreo conceived and designed the experiments, performed the experiments, analyzed the data, wrote the paper, prepared figures and/or tables, reviewed drafts of the paper.
- Rainer Raab and Péter Spakovszky conceived and designed the experiments, collected the feathers, prepared figures and/or tables, reviewed drafts of the paper.
- Juan Carlos Alonso conceived and designed the study and the experiments, contributed reagents/materials/analysis tools, wrote the paper, prepared figures and/or tables, reviewed drafts of the paper.

## Field Study Permissions

The following information was supplied relating to field study approvals (i.e., approving body and any reference numbers):

In Austria, the authors had verbal permission from hunters to collect feathers in their properties. No other official permit is needed in this country to collect feathers. As for Hungary, the authors had a permit to collect gene samples issued for the Hungarian LIFE project (LIFE04 NAT/HU/000109), in which the Hungarian Ministry of Environment and Water was a co-financier and the regional competent Fertő–Hanság National Park Directorate was a partner.

## Data Availability

Raw data is available in the Supplemental Information.

## Supplemental Information

Supplemental information for this article can be found online at http://dx.doi.org/10.7717/peerj.1759#supplemental-information.

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
