# Peer review of "Genetic structure of the threatened West-Pannonian population of Great Bustard (Otis tarda)"

_PeerJ, doi:10.7717/peerj.1759_

## Round 0.1 · original submission · Major Revisions

Please accept my apologies for the late response: most of the reviewers were unresponsive or declined, which delayed the assessment of your article. Said that, after careful consideration, I feel that your work has merit, but is not suitable for publication as it currently stands. Therefore, my decision is "Major Revision."

The two reviews back on your manuscript are positive regarding the science and manuscript content but none addresses the population genetics content in any detail. I have some comments of my own, that I would like to see addressed. Unfortunately, I cannot recommend acceptance at this time. I invite you to submit a revised version of the manuscript that addresses the points raised by both reviewers and my own comments. I encourage you to submit your revision within forty-five days of the date of this decision.

Multivariate analysis. Authors must use DAPC (Discriminant Analysis of Principal Components) to assess population structure, given the different results obtained by Structure and BAPS. DAPC is a multivariate ordination method (Jombart et al., 2008) that does not assume Hardy–Weinberg equilibrium or linkage disequilibrium and is more appropriate for situations where such assumptions are not met, than conventional approaches such as Structure (Pritchard et al., 2000). The advantage of this method that can be run in two different ways: (1) with each individual a priori assigned to its location of origin (a priori population assignment), obtaining for each individual the probability of assignment to their populations of sampling. [This allows to test which of the prior populations an individual could be assigned to best and show whether individuals recently moved from one population to another, indicating admixture.]; (2) using the find.clusters function, determine the number of clusters and assign each individual to a cluster without providing any a priori population assignment. Similar to the a priori population assignment, we obtain a probability of assignment to each cluster for each individual a posteriori (a posteriori population assignment). This removes the effect of assigning populations a priori on the eventual assignment to clusters and offers an unbiased interpretation of population structure.

Microsatellite frequencies. Authors should provide a graph displaying the frequency of alleles (e.g. Teixeira et al., 2012).

Bottleneck. Because the populations in question suffered in the last decades a strong decrease in individuals, I suggest the authors assess the putative genetic bottleneck. That can be done using the software Bottleneck (Piry et al., 1999).

Data availability. There is no reference to the availability of the data. It is now common practice to lodge mtDNA sequences in GenBank, as a population set for instance, with compulsory mention of the GenBank accession numbers for mtDNA sequences. The data files (including microsatellite genotypes) should be made publicly available, either as electronic supplement, or on Dryad repository.


References

JOMBART, T. 2008. adegenet: a R package for the multivariate analysis of genetic markers. Bioinformatics, 24, 1403–1405.
PIRY, S., LUIKART, G. & CORNUET, J. M. 1999. BOTTLENECK: A computer program for detecting recent reductions in the effective population size using allele frequency data. Journal of Heredity, 90, 502—503.

TEIXEIRA S, SERRAO EA, ARNAUD-HAOND S 2012. Panmixia in a Fragmented and Unstable Environment: The Hydrothermal Shrimp Rimicaris exoculata Disperses Extensively along the Mid-Atlantic Ridge. PLoS ONE 7:e38521.

Reviewer 1 ·

Basic reporting

No Comments

Experimental design

No Comments

Validity of the findings

No Comments

Additional comments

The authors present a study wherein they studied the genetic structure of the Austrian population of Great Bustards, a critical issue to better address conservation decision-making. The manuscript is concise and well-written. I particularly appreciated the effort done in the discussion section to put the results in a larger spatial context (confronted with Hungarian and Iberian populations).

I didn’t revise the section on genetic analyses because I only have a basic understanding of these methods. Here are some line-by-line comments on the rest of the manuscript:

L34-43 Could you, if possible, add more precision about time of the recent increase of population, to precise « in the last decades/decade », for instance with the stating dates of management/conservation measures in Austria and Germany.

L38 Add a geographic information (e.g., in Austria) to complete “for the study population” because we have no information at this stage of the introduction.

L65 really in all Austria?

L164-167 The second part of the sentence is confusing, I would say: partly because the intensive protection measures probably stopped early enough the population decline”

L324 reference/source for this presence data?

Reviewer 2 ·

Basic reporting

No comments

Experimental design

No comments

Validity of the findings

No comments

Additional comments

This study reports the genetic diversity, population structure and gene flow of the Great Bustards living in Austria, Slovakia and West Hungary. The authors tested 14 previously reported microsatellite loci and found 12 of them to be useful for their samples. The manuscript is nicely prepared, sampling is sound and the data analysis and interpretation are clear. The findings of this study are novel and help in understanding the current genetic structure of this threatened bird. The authors may include the following relevant studies to discuss the significance of other genetic markers used for Great bustard or other bustard species.
1: Arif IA, Khan HA, Williams JB, Shobrak M, Arif WI. DNA barcodes of Asian Houbara Bustard (Chlamydotis undulata macqueenii). Int J Mol Sci.2012;13(2):2425-38.
2: Riou S, Combreau O, Judas J, Lawrence M, Al Baidani MS, Pitra C. Genetic differentiation among migrant and resident populations of the threatened Asian houbara bustard. J Hered. 2012;103(1):64-70.
3: Idaghdour Y, Broderick D, Korrida A, Chbel F. Mitochondrial control region diversity of the houbara bustard Chlamydotis undulata complex and genetic structure along the Atlantic seaboard of North Africa. Mol Ecol. 2004;13(1):43-54.
4: Pitra C, Lieckfeldt D, Frahnert S, Fickel J. Phylogenetic relationships and ancestral areas of the bustards (Gruiformes: Otididae), inferred from mitochondrial DNA and nuclear intron sequences. Mol Phylogenet Evol. 2002;23(1):63-74.
5: Martín CA, Alonso JC, Alonso J, Pitra C, Lieckfeldt D. Great bustard population structure in central Spain: concordant results from genetic analysis and dispersal study. Proc Biol Sci. 2002;269(1487):119-25.

---

## Round 0.2 · accepted · Accept

Thank you for your constructive responses to my own and the referees' comments and suggestions.

Having read your response and your revised manuscript, I am satisfied that you have considered all the issue, and am happy to accept the manuscript. However, I believe it would be to your benefit to add a DAPC figure (e.g. http://www.sfu.ca/biology/wildberg/NewCWEPage/papers/JonkeretalMolEcol2013.pdf, Figure 3), which can clearly shows the scatter plot of prior and posterior clusters in addition to showing individual membership fraction. However, this is a personal decision which I leave up to you as the authors, and whether you decide to include or not, I am satisfied with your revision and am willing to move the manuscript forward.

Congratulations on the work and thanks for choosing PeerJ.